# Early Steps of Resistance to Targeted Therapies in Non-Small-Cell Lung Cancer

**DOI:** 10.3390/cancers14112613

**Published:** 2022-05-25

**Authors:** Celia Delahaye, Sarah Figarol, Anne Pradines, Gilles Favre, Julien Mazieres, Olivier Calvayrac

**Affiliations:** 1Cancer Research Centre of Toulouse, INSERM UMR1037, CNRS UMR5071, UPS, 31100 Toulouse, France; celia.delahaye@inserm.fr (C.D.); sarah.figarol@inserm.fr (S.F.); pradines.anne@iuct-oncopole.fr (A.P.); favre.gilles@iuct-oncopole.fr (G.F.); 2Laboratory of Medical Biology, Institut Claudius Regaud, Institut Universitaire du Cancer de Toulouse-Oncopole, 31059 Toulouse, France; 3Department of Pharmaceutical Sciences, University Paul Sabatier, 31062 Toulouse, France; 4Department of Pneumology, Toulouse University Hospital, 31059 Toulouse, France

**Keywords:** drug-tolerant persisters, lung cancer, targeted therapies, EGFR-TKI

## Abstract

**Simple Summary:**

Patients with lung cancer benefit from more effective treatments, such as targeted therapies, and the overall survival has increased in the past decade. However, the efficacy of targeted therapies is limited due to the emergence of resistance. Growing evidence suggests that resistances may arise from a small population of drug-tolerant persister (DTP) cells. Understanding the mechanisms underlying DTP survival is therefore crucial to develop therapeutic strategies to prevent the development of resistance. Herein, we propose an overview of the current scientific knowledge about the characterisation of DTP, and summarise the new therapeutic strategies that are tested to target these cells.

**Abstract:**

Lung cancer is the leading cause of cancer-related deaths among men and women worldwide. Epidermal growth factor receptor-tyrosine kinase inhibitors (EGFR-TKIs) are effective therapies for advanced non-small-cell lung cancer (NSCLC) patients harbouring EGFR-activating mutations, but are not curative due to the inevitable emergence of resistances. Recent in vitro studies suggest that resistance to EGFR-TKI may arise from a small population of drug-tolerant persister cells (DTP) through non-genetic reprogramming, by entering a reversible slow-to-non-proliferative state, before developing genetically derived resistances. Deciphering the molecular mechanisms governing the dynamics of the drug-tolerant state is therefore a priority to provide sustainable therapeutic solutions for patients. An increasing number of molecular mechanisms underlying DTP survival are being described, such as chromatin and epigenetic remodelling, the reactivation of anti-apoptotic/survival pathways, metabolic reprogramming, and interactions with their micro-environment. Here, we review and discuss the existing proposed mechanisms involved in the DTP state. We describe their biological features, molecular mechanisms of tolerance, and the therapeutic strategies that are tested to target the DTP.

## 1. Introduction

Nowadays, patients with non-small-cell lung cancer (NSCLC) benefit from more effective and personalised treatments, as demonstrated by the continuous improvement in overall survival of cancer patients [1,2]. In recent decades, high-throughput sequencing has enabled the identification of genetic alterations on oncogenic drivers (Figure 1A) and led to the development of targeted therapies (Figure 1B) that greatly improved response rates and overall survival [3,4]. Epidermal growth factor receptor-tyrosine kinase inhibitors (EGFR-TKIs) are one of the first examples of this therapeutic breakthrough, as they provide a greater and more sustained response in EGFR-mutated lung adenocarcinoma patients than any other existing anti-cancer therapies [5], including immunotherapies [6,7].

*EGFR* gene encodes the epidermal growth factor receptor (EGFR; ErbB-1; HER1) and belongs to the ERBB family of tyrosine kinase receptors [8]. EGFR is broadly expressed across normal tissues [9] and controls cell survival and proliferation pathways [10]. In NSCLC, EGFR-activating mutations are detected in 10 to 40% of patients with disparities between continents and countries [11,12] and are mostly located in exons 18 to 21 that notably encode the ATP-binding pocket of the receptor tyrosine kinase domain [13]. These alterations are more frequently found in non-smoking patients (44%) and women (21%) [3].

*EGFR* mutations were discovered in 2004 following the genotyping of tumours from a subset of patients who were highly responsive to first-generation inhibitors erlotinib and gefitinib [14,15]. Subsequent clinical trials confirmed the efficacy of these treatments for patients harbouring *EGFR*-mutated adenocarcinomas [5,16,17,18,19,20,21]. However, despite a strong initial response, patients invariably experienced recurrence after a median of one year of treatment. Most resistance mechanisms to first-generation EGFR-TKI have been identified, in particular the *EGFR*-T790M secondary mutation for half of the patients [22,23] (Figure 2), which led to the development of second-generation inhibitors, such as afatinib [24] and dacomitinib [25], which demonstrated selectivity to *EGFR*-T790M, but also a poor benefit/toxicity ratio. The second-generation inhibitors were therefore quickly supplanted by the third-generation EGFR-TKI, osimertinib [26] and rociletinib [27] (Figure 2). These inhibitors were first developed in patients who became resistant to first- and second-generation EGFR-TKI [28,29], and osimertinib is now approved as first-line therapy in patients with activating *EGFR* mutations [30,31]. Nevertheless, resistance to these third-generation inhibitors are also reported, with, for example, the acquisition of C797S tertiary mutation on *EGFR* that prevents the binding of the inhibitor [32] (Figure 2). Recently, allosteric EGFR inhibitors have been developed to overcome resistance mediated by classical ATP-competitive compounds [33] (Figure 2), which display encouraging preclinical results that require clinical validations.

Resistance mechanisms to EGFR-TKI can be classified in three main groups [34]: (1) Target alterations: secondary mutations on *EGFR* or *EGFR* amplifications [35]; (2) Activation of bypass signalling pathways, such as PI3K/AKT, MAP kinase pathway, IGFR1, AXL, IGFR1/KDM5A, or amplifications of *MET*, *HER2* or *SMO* [36,37,38,39,40,41]; and (3) Transformation of the cellular phenotype by epithelial to mesenchymal transition (EMT) or histological transformation to small-cell lung cancer (SCLC) [22,42,43,44]. All generations of EGFR-TKI lead to these three main types of acquired resistance, only their proportions differ from one inhibitor to another [45] (Figure 2).

This never-ending emergence of the same resistance mechanisms, despite improvements of the successive generations of inhibitors, illustrates how therapeutic interventions are constantly pushing cancer cells to find new ways to adapt and escape treatments. This detrimental dynamic relies on cancer cell plasticity and selective pressure on tumour cells to adapt. This observation questions the therapeutic strategy that consists of targeting resistance at the time of relapse, and suggests that a more effective strategy should rather target the dynamic mechanisms that led to the acquisition of resistance. It is therefore crucial to understand the origin of the emergence of resistance mechanisms to offer new therapeutic strategies to prevent or delay relapse. Two hypotheses could explain the acquisition of resistance mechanisms to EGFR-TKI [46] (Figure 3): (1) the Darwinian selection of a genetic or non-genetic pre-existing resistant subclone; and (2) the Lamarckian model suggesting that the resistance mechanism is acquired during treatment from a subpopulation of cells called “Drug-Tolerant Persisters” (DTPs). In this hypothesis, DTPs overcome treatment-induced cell death by entering a transient and reversible pseudo-dormant state [47] before developing de novo resistance mechanisms conferring a proliferative advantage, thus causing relapse [46,48]. Both mechanisms of adaptation have been found to co-exist within a same-cell population [46,49,50] and may also be clinically relevant. While the first hypothesis might be a relevant model for the observation of primary or early resistance in patients, where tumours do not initially respond to treatment, the second hypothesis may explain the relapse in patients that initially presented a durable response to treatment. (Figure 3) Nevertheless, no information is currently available to assess the existence of each of these processes in patients.

An increasing number of studies have been performed to describe the dynamic process of resistance acquisition upon treatment, using various models. In this review, we describe the biological features of DTPs, the mechanisms of tolerance used by these cells, and the therapeutic strategies that have been proposed to target them. We focus on *EGFR*-mutated non-small-cell lung cancer treated with EGFR-TKI, but we also describe similarities in other oncogenic contexts upon targeted therapies or chemotherapies.

## 2. Biological Features of Drug-Tolerant Persisters (DTPs)

DTPs have been first described in bacteria. Antibiotic-tolerant bacteria survive under antibiotic treatment and are able, after antibiotic withdrawal, to regain proliferative capacities as well as antibiotic sensitivity [51,52,53,54]. This indicates that phenotypic plasticity of antibiotic-tolerant bacteria upon treatment is not mediated by genetic alterations. In cancer, these DTPs were first described in 2010 by Sharma et al. in the *EGFR*-mutant NSCLC PC9 cell line, where authors observed a small fraction of largely non-proliferating cells that resisted to EGFR-TKI treatments at doses 100 times higher than the IC50 value [47]. Some of these DTP could resume proliferation in the presence of the inhibitor and were referred to as drug-tolerant expanded persisters (DTEPs) [47]. Since then, many studies have focused on characterising the DTP population (Figure 3), mainly by using the PC9 cell line as a preferred study model, of which the phenotypic features are discussed in this section (Figure 4).

### 2.1. Reversible Pseudo-Dormant Phenotype

Sharma et al. showed that DTPs represented 0.3% of the initial PC9 cell population nine days after erlotinib treatment. They were described as slow-cycling cells, as about 75% of PC9 EGFR-TKI-induced DTPs were arrested in the G1 phase of the cell cycle [46,47,55]. Similarly, HCC4006 cell-line-derived xenograft tumours treated with osimertinib showed a high decrease in Ki67-positive cells [56]. Thus, the drug-tolerant state is often referred to as dormant or quiescent [57]. However, approximately 20% of erlotinib-induced PC9 DTP were able to recover normal proliferation in the presence of the drug, and could then proliferate indefinitely under treatment to form DTEP clones [47]. Ramirez et al. later showed that DTEP could harbour a large diversity of EGFR-TKI-resistance mechanisms similar to those found in patients, such as *EGFR*-T790M mutation, *MET* amplification, and alterations in the MAP kinase and the PI3K/AKT pathways [48]. This suggests that the slow-cycling population of DTPs may act as a reservoir of cells gradually acquiring drug-resistance mechanisms and later giving rise to fully proliferative clones. Interestingly, it has been shown in vitro that a period of drug holiday resulted in a loss of the resistance mechanism and a re-sensitisation of tumour cells [47,58,59], highlighting the reversibility of the resistance mechanism, which seems to be dependent on the drug-selection pressure. In patients, the re-sensitisation of the tumours after targeted therapy interruption have also been reported [22,60,61], although drug discontinuation does not seem to be a valid therapeutic option as patients experience tumour regrowth and accelerated disease progression [62].

A similar pseudo-dormant and reversible drug-tolerant phenotype have also been reported in other oncogenic contexts, such as glioblastoma [63,64], *BRAF*-mutant melanoma [65,66,67], colorectal cancers [59,68,69], breast cancers [70], or pancreatic cancer [71], treated either with targeted therapies or chemotherapies. This suggests that the drug-tolerant state is not specific to a particular type of cancer and/or therapy, although the intrinsic phenotypic characteristics of DTC may differ between to the different models.

Recent studies are in favour of a heterogeneous and dynamic state, rather than a stable dormant state. Indeed, Oren et al. demonstrated, using a high-complexity lentiviral barcoding strategy, that the DTP population from the PC9 cell line or from other oncogenic contexts contains a rare proliferative subpopulation that cycle early in the course of treatment with osimertinib and other targeted therapies [72]. Echeverria et al. also demonstrated that residual breast cancer tumours under chemotherapies have a Ki67-positive cycling cell subpopulation [73].

### 2.2. Senescent-Like Phenotype

Consistent with a non-proliferative phenotype, senescence characteristics have been observed in EGFR-TKI-derived DTPs. Indeed, EGFR-mutant cell lines have been described to express some senescence-associated markers in response to EGFR-TKI, such as senescence-associated β-galactosidase (SA-βGal) activity, senescence-associated secretory phenotype (SASP), an increase in cell size, upregulation of gene signatures associated with senescence, increased H3K9me3-positive nuclei foci, and the induction of the cyclin-dependent kinases inhibitors (p16, p21, p53, and p27^Kip1^) [74,75]. Therapy-induced senescence (TIS) has also been demonstrated under targeted therapies or chemotherapies in many others cancers, again highlighting the comparable phenotypes of tolerance despite different therapeutics [76,77,78,79,80,81,82,83,84,85] and is referred to as “pseudo-senescence”, “senescence-like arrest”, or “senescent-like phenotype” [86]. However, while senescence was thought to be an irreversible phenomenon leading to cell death, some studies suggest that, under therapy, cells are able to escape from this state of senescence to regain proliferative capacities and cause tumour relapse. In 2018, Milanovic et al. showed, in haematological cancers models, that TIS results in profound genetic reprogramming that promotes stemness and escape from cell cycle arrest that can lead to tumour relapse. Furthermore, tumour cells that have escaped TIS are more malignant and aggressive, as they have increased tumour-initiation capabilities in immunocompetent mouse models compared to cells that have never been senescent [83]. However, it is unclear whether the proliferative clones that emerge during treatment are directly derived from cells with senescent features or from a cell subpopulation that does not express senescence features. Saleh et al. reported that lung, colorectal, and breast cancer cell lines induced into senescence by etoposide or doxorubicin treatments are able to recover their proliferative capabilities under treatment. Moreover, this recovery of proliferative capacity is found for the total cell population, but also for the population of cells enriched in senescent phenotypes. This recovery of proliferation of cells enriched in the senescence phenotype is demonstrated by the observation of mitoses parallel to the progressive loss of senescent characteristics—SA-βGal activity and cell size. Furthermore, cells enriched in the senescent phenotype are able to form tumours after xenografts in immunodeficient and immunocompetent mice [87]. In vivo, in transgenic mouse models, Demaria et al. demonstrated that chemotherapy-induced senescence promotes tumour relapse and chemotherapy-related adverse events [88]. Recently, Duy et al. demonstrated that acute myeloid leukaemia (AML) cells enter a senescence-like phenotype following chemotherapy in vitro and in vivo on PDX models. This senescent phenotype confers superior colony-forming and engraftment potential. Senescence entry is dependent on ATR and ATR inhibitors are able to impede AML persistence [77]. To date, escape from EGFR-TKI-induced senescence has not been clearly demonstrated and incriminated as a novel mechanism of resistance to EGFR-TKI. More studies are needed in order to decipher the role of this pseudo-senescent phenotype in drug tolerance and resistance.

### 2.3. Phenotypic Plasticity

In 2–15% of the cases, the mechanism of resistance to EGFR-TKI in NSCLC involves a histological transformation to small-cell lung cancer (SCLC) [22,42,43], highlighting the cancer cell’s plasticity and ability to phenotypically adapt to survive drug pressure. Moreover, several studies have shown that NSCLC models acquire resistance through EMT [89,90,91]. EMT is a biological process by which an epithelial cell undergoes molecular and biochemical changes resulting in a mesenchymal phenotype, and is a central process in tumour progression, metastasis, and anti-cancer drug resistance [92,93]. In EGFR-TKI-induced DTPs, transcriptional profiles by RNA-seq experiments demonstrated the upregulation of EMT-gene signatures compared to untreated control cells [46,75,94,95]. EMT signatures have also been correlated with DTP states in other models under targeted therapies or chemotherapies [70,96,97,98]. Nevertheless, the specific markers of the mesenchymal phenotype, such as N-cadherin and Vimentin, or the overexpression of transcriptional factors involved in EMT, such as ZEB1/2, SNAIL, and SLUG, are more often observed in resistant cells and have not been observed in DTPs at the protein level. Therefore, further studies are required to deepen the connection between the EMT process and anti-cancer drug tolerance, in order to ensure that the upregulation of the EMT signature observed in DTPs is recapitulated at the protein level, as well as to prove that the presence of EMT markers in some DTPs is not the reflection of early resistance.

Several studies showed that DTPs in *EGFR*-mutated models treated with EGFR-TKI express cancer stem cell markers. Cancer stem cells are able to perpetuate themselves through self-renewal and to generate differentiated cells. Stemness properties can influence cancer initiation, cancer progression, metastatic growth, but also therapy resistance [99]. Indeed, two studies showed that EGFR-TKI DTPs in NSCLC models overexpress stemness markers CD133, CD24, and ALDH [47,100], and harbour a sphere-forming ability and clonogenic potential [101]. Kunimasa et al. demonstrated that DTPs induced by gefitinib treatment overexpressed OCT3/4, NANOG, SOX2, CD133, and ALDH1A1A [74]. Expression of stemness markers has also been reported in various cancers under therapeutic pressure: JARID1B [67] and CD271 [66] in melanoma; SOX2, SOX4, NFIA, and OLIG2 in glioblastomas [63,64]; stemness gene signature in AML [77]; CD133 and CD44 in breast cancers [73,97,102]; CD24, CD44, CD133, CXCR4, ALDH1A3, higher ALDH activity, and higher sphere-forming ability in colon cancers [69]; CD133, CD44, and ALDH activity in KRAS-ablation of pancreatic cancer models [71]. However, the expression of stemness markers seems to be very heterogeneous in DTP population [73,74,77] and stem-like phenotype involvement in tolerance need to be confirmed by functional studies.

The phenotypic plasticity of DTPs has also been described in vivo and in patients. Using melanoma PDX samples, Rambow et al. showed that dedifferentiation of melanoma cells to a transient neural crest stem cell population contributes to the development of resistance to targeted therapy [103,104]. In a recent clinical study, Maynard et al. performed scRNA-seq on advanced-stage NSCLC at different stages of treatment (untreated, residual disease, and progressive disease). The transcriptional profiles of the samples showed a significant increase in alveolar cell-specific gene expression in residual disease samples compared to untreated samples. They correlated a high expression of their alveolar signature in tumours with improved patient overall survival. They suggest that the expression of alveolar genes could lead to repair and escape from cell death, leading to treatment persistence and less aggressive tumours, while constituting a pool of cells that could acquire genetically driven resistance [105]. Altogether, these data suggest that treatment induces a phenotypic transition through differentiation.

## 3. Mechanisms of Tolerance

Mechanisms of tolerance that enable DTPs to survive upon treatment are described in this section and illustrated in Figure 4.

### 3.1. Chromatin Remodelling and Epigenetic Modifications

Epigenetic modifications are defined as the changes in gene function that do not entail a change in DNA sequence [106,107]. Epigenetic mechanisms coordinate chromatin structure and regulatory proteins—transcription factors—to specifically activate or repress gene expression. The epigenetic modifications involve histone modifications, DNA methylation, nucleosome remodelling, and non-coding RNAs. They alter chromatin structure and serve as docking sites for specialised proteins that read the modifications and translate their information through the protein effectors, which altogether play a key role in transcription, DNA repair, and replication [108]. Epigenetic regulatory mechanisms are essential in cell-fate specification during development, and more broadly allow cells to respond and adapt to environmental changes. Consequently, the alteration of the epigenome has critical consequences on the cell biological processes and is a hallmark of cancer, as well as a characteristic of the DTP population as described in recent studies.

Two biological features of the DTP population, described in the first section of this review, indeed strongly suggest that the mechanisms of acquisition of resistance could involve epigenetic modulation. First, the DTP state is reversible, as shown by Sharma et al., where DTPs cultured in drug-free media reacquired proliferation capacities and EGFR-TKI sensitivity within a short period of time [47]. This observation is in favour of a transient non-genetic state, hence likely to go through epigenetic modifications. Second, phenotypic plasticity is also a major characteristic of DTPs, and is linked to epigenetic reprogramming. At the transcriptional level, droplet-sequencing (DROP-seq) experiments emphasised the plasticity of drug tolerance, as the signature of drug-tolerant markers was lost upon drug withdrawal, and regained quickly after drug re-sensitisation [94]. The same study highlighted the involvement of epigenetic regulation in the DTP state, by identifying genes associated with EGFR-TKI-resistant cell populations, including epigenetic regulation effectors [94].

DNA methylation is the most extensively characterised chromatin modification. DNA methylation is mostly located on centromeres, telomeres, inactive X-chromosome and repeat sequences. In malignant cells, global hypomethylation is commonly observed, particularly on repeated elements, which leads to genomic instability [109], with the exception of cytosine-guanine (CpG) islands that are frequently hypermethylated. CpG islands are present in 70% of all mammalian promoters and are indeed abnormally methylated in various cancer genomes, affecting the expression of protein-coding genes and non-coding RNAs [108,110]. In the context of drug tolerance, the differential expression of DNA methyltransferases has been reported in DTPs compared to control cells in various cancers (lung, melanoma, breast, and colon cancers) treated with chemotherapeutic agents or targeted therapies [111]. However, genome-wide DNA methylation analyses did not reveal a common pattern in DTP cell lines, but rather that each cell line followed a different trend of methylation level depending on its cancer type. This suggests that DNA methylation re-arrangement in DTPs is a cell-type specific response and that epigenetic modulation in DTPs is driven by histone modifications [111].

Several studies on DTPs have highlighted a common epigenetic reprogramming signature mainly guided by histone modifications with a global decrease in H3K4me3 and an increase in H3K9me3 levels. Sharma et al. first suggested that a global chromatin remodelling event must occur in DTP cells, as they observed differential nuclease sensitivity in parental and DTEP cells in lung and colorectal cell lines. They reported a general decrease in H3K4me2/3 in DTP lines that is consistent with the upregulation of the H3K4 demethylase KDM5A [47]. Methylation on H3K4 is mostly associated with active or permissive chromatin regions and is found on transcription start sites of active genes [112]. They later confirmed these observations with mass spectrometry on histone H3 post-translational modifications (PTMs) [113]. Assay for transposable-accessible chromatin followed by sequencing (ATAC-seq) experiments demonstrated a global chromatin remodelling in DTP cells compared to parental lines [75,113]. Other studies also identified the upregulation of KDM5A and KDM5B in DTPs, and their inhibition by genetic knockdown or chemical inhibitors decreased the number of DTPs and subsequent resistant clones in several cell lines, indicating the requirement of the chromatin modifiers to establish a drug-tolerant state [66,67,114,115,116].

Methylation of H3K9 is largely associated with gene silencing and is a characteristic hallmark of repressive heterochromatin. It is also an established marker of cell senescence [117] and is part of the epigenetic signature shared between several cancer cell lines DTPs [66,75,111,113]. Guler et al. reported that DTPs exhibited a repressed chromatin state consistent with overexpression of K9 methyltransferases, but, surprisingly, this did not correlate with a general decrease in the expression of protein-coding genes. Instead, they found H3K9me3 accumulation over LINE-1 elements in particular. Knockdown and knockout approaches, as well as H3K9 methyltransferase-inhibitor treatment, decreased the number of DTPs, suggesting that the H3K9me3-mediated repressive state is crucial for DTP survival [113].

In line with a repressive state, lung cancer DTPs display a global decrease in H3K4 acetylation. Acetylation is associated with active genes as it neutralises the positive charge of lysine residue, weakening the electrostatic interaction between negatively charge DNA and histones, thus disrupting the tight packaging of chromatin. Treatment with histone deacetylase (HDAC) inhibitors in combination with EGFR-TKI indeed strongly impedes the establishment of lung cancer DTPs [47,113]. This demonstrates that the mechanisms underlying drug tolerance are dependent on HDACs’ expression to maintain a global repressive state.

Importantly, the histone PTM specific to DTP stages are lost when DTPs are allowed to re-grow following drug removal [75,113], which is consistent with the transient phenotype of DTPs. Epigenetic modifications are, by definition, dynamic and reversible, and are thus likely to be key players in the adaptation of the cells following drug exposure by contributing to transcriptional reprogramming and cell differentiation. However, the mechanisms underlying the alterations of histone PTM in the DTP population still remain to be elucidated, as well as the mechanisms that drive the acquisition of resistance following these epigenetics modifications.

### 3.2. Resistance to Cell Death and Cell Signalling Reprograming

Escape from cell death is one of the hallmarks of cancer and is a common cause for resistance to treatment via various mechanisms, including the escape from apoptosis and cell signalling reprogramming.

#### 3.2.1. Escape from Apoptosis

Apoptosis is the main process of programmed cell death; it plays a critical role in maintaining tissue homeostasis and is frequently dysregulated in different types of cancer. Apoptosis is an active process mediated by extrinsic and intrinsic pathways that both activate cysteine-aspartic proteases, known as caspases, ultimately resulting into the controlled death of the cell. The extrinsic pathway is initiated by the binding of members of the tumour necrosis factor (TNF) superfamily to their cell-surface death receptors, while the intrinsic pathway is the result of internal cellular stresses activating the B-cell lymphoma (BCL-2) family. Resistance to apoptosis is a frequent cause of resistance to targeted therapies, including for primary resistance. Two studies in lung cancers showed that a low expression of BIM, a member of the BCL-2 family, in treatment-naïve patients prevents the induction of apoptosis and mediates an intrinsic resistance to EGFR-TKI [118,119].

Not surprisingly, the persistence of DTPs following drug exposure is directly linked to their capacities of escaping cell death to survive. In NSCLC models, Hata et al. proposed that late-emerging *EGFR* T790M-acquired resistance results from the evolution of drug-tolerant cells, and showed that these cells displayed a reduced apoptotic response during EGFR-TKI treatment. This was also true for patient-derived cell lines, where the least osimertinib-sensitive cells exhibited lower apoptotic responses. This suggests that these cells present a decreased dependence on EGFR activation for survival. Co-treatment with navitoclax, a dual BCL-x and BCL-2 inhibitor, significantly increased apoptosis in cell lines and xenografts [46]. More recently, Kurppa et al. generated DTPs following the combined treatment of osimertinib and trametinib, an MEK inhibitor. They showed that the establishment of the DTP was critically dependent on the activation of YAP/TEAD, and that the complex YAP/TEAD/SLUG mediates the repression of the pro-apoptotic factor BMF. Pharmacological inhibition of YAP/TEAD led to a robust increase in BMF level and thus to an increase in apoptosis [75]. YAP1 has also been reported to mediate survival in a study by Tsuji et al. They showed that YAP1 was activated in alectinib-treated ALK-rearranged lung cancer cells and bound to MCL1 and BCLXL upstream regions, thus increasing the expression of the two anti-apoptotic genes [120]. In *EGFR*-mutated NSCLC cell lines treated with EGFR-TKI, another work linked the EMT transcription factor TWIST1 with the repression the pro-apoptotic factor BIM. They showed that TWIST1 mediates the repression of BCL2L11 gene encoding for BIM by binding upstream of its transcriptional start site and within an intron. The downregulation of TWIST1 increased BIM protein level and resulted in increased sensitivity to EGFR-TKI [121]. Finally, UFMylation, the process by which ubiquitin-fold modifier 1 (UFM1) is conjugated to its target proteins, also seems to be involved in DTP survival. Its inhibition resulted in a protective unfolded protein response pathway, promoting BCL-x dependency [122].

#### 3.2.2. Cell Signalling Reprogramming

Cell signalling reprogramming and the rearrangement of the gene-regulatory network is a mechanism of cell survival in response to treatment. Various mechanisms are involved in drug tolerance; the relevant pathways committed to DTP establishment and maintenance in NSCLC are briefly described below.

Despite the low proliferative capacities of DTPs and their slow-cycling phenotype, a few cell cycle regulators have been efficiently targeted to delay acquired resistance. Targeting transcription and cell cycle regulators has been tested in PC9 lung cancer cell lines, in particular CDK7, which is a key regulator of cell cycle and transcription initiation. Rusan et al. co-treated the cells with erlotinib and THZ1, a CDK7 inhibitor, leading to increase cell death and preventing the emergence of resistance in vitro [123]. A study by Shah et al. also demonstrated the role of Aurora kinase A (AURKA) in the establishment and maintenance of DTPs. AURKA regulates chromosome alignment, mitotic spindle formation, and chromosome segregation during the G2/M phase of the cell cycle. In lung cancer model, its activation is sufficient to cause resistance to EGFR-TKI, while its chemical inhibition reduces cell proliferation under EGFR-TKI and increases apoptosis in vitro and in a PDX model [124].

AXL protein activation has been extensively implicated in EGFR-TKI resistance [41]. AXL is a receptor tyrosine kinase involved in cell survival, proliferation and migration, immune response and inflammation, by activating PI3K/AKT/mTOR, MERK/ERK, NFκB, and JAK/STAT pathways [125,126]. AXL upregulation, together with its ligand GAS6, was observed in 20% of first-generation EGFR-TKI-resistant patients. Moreover, AXL inhibition enables re-sensitisation to erlotinib and a co-treatment with EGFR-TKI and AXL inhibitor delays the emergence of resistance in vitro and in vivo [41,55,127,128,129].

IGF1R activation has been reported to mediate DTP establishment in NSCLC. For instance, its activation together with the downregulation of IGFBP3/4 bypass gefitinib-mediated EGFR inhibition by activating the PI3K/AKT pathway, whereas its inhibition allows cells to be re-sensitised to gefitinib [130]. Sharma et al. also evidenced that the establishment and viability of lung cancer DTPs are dependent on IGF1R overexpression and phosphorylation, mediated by the upregulation of the histone demethylase KDM5A [47].

The activation of PI3K/AKT and MEK/ERK pathways also occurs via the activation of the FGF2/FGFR1 pathway under EGFR-TKI treatment, and promotes EMT-like transcriptional changes and cell survival. Overexpression of FGF2 and FGFR1 was reported in lung cancer cell lines early on, following targeted therapies [95,122,131,132].

Phuchareon et al. worked on gefitinib-induced DTPs and observed that AKT inhibition leads to a downregulation of the transcription factor ETS1, thus decreasing the expression of its target genes: cyclins D1, D3, and E2. This is consistent with the pseudo-dormant phenotype described in DTPs. The phosphatase DUSP6 is also downregulated due to the decreased expression of Ets-1. DUSP6 is a negative regulator of ERK1/2, hence its silencing induces the reactivation of ERK1/2 and the subsequent MAP kinases pathway activation, responsible for cell survival [36,133]. Concordantly, MEK inhibitors associated with EGFR-TKI prevents the emergence of various acquired resistance in vitro and in vivo [134], but does not fully eradicate them, as Kurppa et al. obtained a DTP population following a co-treatment with EGFR-TKI and trametinib, an MEK inhibitor. As described in the previous section (escape from apoptosis), these cells harboured an increased dependence on YAP/TEAD activity, which is involved in cell proliferation and apoptosis/survival balance.

The notch-signalling pathway seems to be one of the EGFR-TKI-tolerance mechanisms in lung cancer. Notch signalling is involved in a variety of biological processes, such as differentiation, maintenance of stemness, apoptosis, cell survival, and growth arrest [135]. Its inhibition was shown to re-sensitise human and murine lung adenocarcinoma resistant to gefitinib, as well as human cell lines harbouring the osimertinib driver resistant mutation *EGFR* C797S, via the inhibition of HES1 protein by the binding of phospho-STAT3 to its promoter [136]. Another study shows that targeted therapies induce a non-canonical activation of Notch3 that leads to b-catenin stabilisation and promote cell survival [101]. In patient biopsies, Maynard et al. also showed that beta-catenin pathway-specific genes are enriched in the residual disease stage compared to untreated tumours, together with alveolar gene-expression signature. The WNT/b-catenin pathway is involved in oncogenesis, repair, and regeneration after cell injury, and might therefore contribute to the establishment of the drug-tolerance state in vivo [105]. Interestingly, the chemical inhibition of the WNT/b-catenin pathway increased the sensitivity to EGFR-TKI in vitro [75,105].

### 3.3. Metabolic Reprogramming

Cellular metabolism is fundamental for supporting cell activities, including survival and proliferation [137] in normal or tumourigenesis contexts [138]. A growing number of studies show that metabolic reprograming plays a role in response and adaptation to therapies [139].

#### 3.3.1. Mitochondrial Respiration

DTPs need to increase energy production to survive and adapt to therapeutic pressure. To produce energy, DTPs shift their metabolism from glucose consumption by glycolysis to mitochondrial oxidative respiration. For example, EGFR-TKI treatment suppresses glycolysis in parental *EGFR*-mutant lung adenocarcinoma lines and induces a dependence on mitochondrial oxidative phosphorylation (OxPhos) for cell survival. Consistently, it was shown that OxPhos inhibitors significantly delay the development of osimertinib resistance [140,141]. In pancreas KRAS-ablation-resistant models, Viale et al. showed that DTPs have more active mitochondria and the upregulation of OxPhos, but a decreased glycolysis. Inhibition of OxPhos in these KRAS-ablation models leads to prevent tumour recurrence [71]. Echeverria et al. demonstrated that residual tumours of breast cancer treated with chemotherapy are dependent on mitochondrial OxPhos and using an inhibitor of respiratory chain complex I leads to delayed tumour relapse [73]. In similar models, Goldman et al. showed that DTPs have an increased mitochondrial respiration and ROS production, as well as an increased glycolysis process [102]. Roesch et al. showed that DTPs in a BRAF-mutated model treated with targeted therapy and chemotherapy have an upregulation of mitochondrial OxPhos and inhibition of ATP synthase or respiratory chain complex I allows to restore sensitivity to drugs [67]. In BRAF-mutated multiple myeloma under BRAF/MEK inhibitor, DTPs also show an upregulation of OxPhos [142].

Lipid metabolism can also be used to produce energy for mitochondrial respiration and DTP survival. Indeed, DTPs have an upregulation of fatty acid oxidation (FAO) to provide substrate for the mitochondrial respiratory chain. Oren et al. showed that cycling and non-cycling persisters have distinct transcriptional and metabolic programs. Proliferative persisters have antioxidant gene upregulation and a metabolic reprogramming to FAO. Impeding oxidative stress or metabolic shift leads to reducing the fraction of persister cells [72]. Aloia et al. demonstrated that DTPs in BRAF-mutated models treated with targeted therapy have an upregulation of the fatty acid transporter CD36 and of FAO process. FAO inhibition induces the upregulation of glycolysis, and glycolysis associated with MAPK inhibition delays tumour relapse [143]. Shen et al. verified these observations and showed that persisters of BRAF-mutated melanoma treated with BRAF/MEK inhibitors proceed to a metabolic switch from glycolysis to OxPhos supported by peroxisomal FAO and peroxisomal FAO inhibitor triggers the reduction in the emergence of persister cells [144]. Feng et al. also showed that CD36 is upregulated in breast cancer resistant to HER2-targeted therapy and CD36 inhibition sensitises resistant cells to the drug [145]. In acute myeloid leukaemia models upon chemotherapy, Farge et al. demonstrated that persister cells exhibit an upregulation of ROS, OxPhos, and FAO and targeting this metabolic shift allows an increase in chemotherapy efficacy [146]. In BCR-ABL chronic myeloid leukaemia treated with imatinib, Kuntz et al. also showed that leukemic residual stem cells have an upregulation of oxidative metabolism and inhibition of mitochondrial protein translation allows the eradication of minimal residual disease in vitro and in vivo [147].

In order to produce energy, persister cells also need substrates. Guo et al. demonstrated that the autophagy process allows the recycling of substrates, such as amino acids and nucleosides required for mitochondrial metabolism via TCA (tricarboxylic acid) cycle [148]. In melanoma treated with BRAF inhibitors, it has been demonstrated that autophagy is induced and is responsible for drug resistance [149,150].

#### 3.3.2. Oxidative Stress

In DTPs, overactivation of mitochondria metabolism leads to an increase in oxidative stress with reactive oxygen species (ROS) production, and several studies have shown indeed that DTP survival depends on anti-oxidative factors. Glutathion-dependent reduction in lipid peroxides is one of the major anti-oxidant processes to reduce oxidative stress. Viswanathan el al. demonstrated that DTPs have an upregulation of polyunsaturated lipid synthesis and that the lipid-peroxidase GPX4 pathway is crucial for DTP survival by interfering with ferroptosis-mediated cell death [90]. Inhibition of GPX4 in DTPs of breast cancer, lung cancer, and melanoma allows the induction of ferroptotic death of DTPs in vitro and to prevent relapse in mice models [97]. Similarly, the inhibition of the antioxidant activity of ALDH in EGFR-TKI-induced DTPs led to the accumulation of ROS sufficient to induce cell death [100]. Wang et al. demonstrated that EGFR-TKI-induced DTPs have an upregulation of BCAT1 (branched-chain amino acid aminotransferase I) mediated by H3K9 demethylation. BCAT1 has an anti-oxidant activity by reducing ROS. Thus, EGFR-TKI combination with ROS-inducing agents allows to overcome resistance [151,152]. More recently, transcriptional analysis of early cycling cells in EGFR-TKI-induced DTP revealed that they exhibited a higher expression of glutathione metabolism and NRF2 signatures than non-cycling persisters [72]. The fraction of cycling cells in the DTP population was increased when they alleviated ROS by treatment with the ROS scavenger N-acetylcysteine (NAC), but decreased when they impeded glutathione synthesis, suggesting a strong relationship between antioxidant species signatures and persister proliferative capacity [72]. Russo et al. showed in colorectal cancer cells that anti-BRAF- and anti-EGFR-targeted therapies induce ROS production and DNA damage leading to mutagenesis in persister cells [59]. Fox et al. also showed that, upon HER2 inhibition in breast cancer models, DTPs have an enhanced oxidative stress with upregulation of the antioxidant NRF2, which promote de novo nucleotide synthesis and relapse. In this context, inhibition of glutaminase prevents DTP reactivation and tumour relapse [153]. Zhang et al. also reported that, in prostate and breast cancer models, chemotherapy-induced DTPs have an increased oxidative stress that can be prevented by scavenging anti-oxidant NRF2-NPC1L1 pathway, which regulate vitamin E uptake [154]. Thus, mechanisms of metabolic reprogramming leading to DTP survival are plentiful and involve oxidative stress, FAO, and OxPhos processes. Targeting these mechanisms represents opportunities to target and selectively eradicate DTPs [155].

### 3.4. Microenvironment Implications

Solid tumours contain cancer cells but also stromal cells, including fibroblasts, immune cells, endothelial cells, as well as blood vessels and an extracellular matrix. This complex tumour micro-environment (TME) can regulate tumour progression, metastasis, but also therapeutic response [156,157,158]. In this paragraph, although the mechanisms we describe are more related to drug resistance, they also share some similarity to drug tolerance.

It has been demonstrated that cancer-associated fibroblasts (CAFs) are able to promote resistance to targeted therapies. For example, CAFs can secrete Hepatocyte Growth Factor (HGF), which activates his receptor, HGRF or c-Met, and the PI3K/AKT downstream pathway leading to drug resistance. Indeed, CAFs are able to induce the EMT process, and therefore resistance in *EGFR*-mutated NSCLC cells upon EGFR-TKI by secretion of HGF and Insulin Growth Factor 1 (IGF-1) [159,160]. The same observations have been made in BRAF-mutated melanoma, glioblastoma, and colorectal cancers upon BRAF inhibitors [161]. HGF secretion by CAFs is supported by lactate hypersecretion by cancer cells, which demonstrates a real dialog between CAFs and cancer cells [162]. Moreover, Wilson et al. showed that HGF promotes resistance to lapatinib in HER2-amplified breast cancers cells and to BRAF inhibitor in BRAF-mutated melanoma cells [163]. Hirata et al. also demonstrated that CAFs induce an activation of Integrin β1/FAK (Focal Adhesion Kinase)/Src signalling leading to resistance to BRAF inhibition in melanomas in vivo [164]. CAFs are also involved in resistance to chemotherapies. Indeed, Zhang et al. demonstrated that CAFs can induce resistance to several chemotherapies by the secretion of IGF-2 that induces the AKT/Sox2 pathway in NSCLC [165]. In lung and breast cancers, Su et al. showed that a specific subset of CAFs (CD10+, GPR77+) can promote chemoresistance by supporting cancer stem cell phenotype via IL-6 and IL-8 secretion [166]. In pancreatic cancers, CAFs expressing IRAK4 can induce resistance to chemotherapy by IL1β secretion that activate the NFκB pathway in pancreatic cancer cells [167]. In colorectal cancers, CAFs induce stemness and EMT of cancers cells by exosomal miR-92a-3p secretion leading to resistance to chemotherapy [168]. Exosomal miRNA secretion by CAFs has also been involved in ovarian cancer chemoresistance. Indeed, Au Yeung et al. demonstrated that CAFs and cancer-associated adipocyte (CAA) exosomes are able to deliver miR21 to cancer cells and to induce resistance to paclitaxel by binding to his target, AFAP1 [169].

Tumour-associated macrophages (TAMs) can also be involved in resistance to anticancer therapies [170]. In breast cancer, HER2 inhibition leads to TNFα/NFκB pathway activation and the subsequent secretion of inflammatory cytokines, including CCL5. CCL5 can recruit CCR5-expressing macrophages in the tumour leading to cancer recurrence [171]. Similarly, in prostatic cancer, TAM can release CCL5 leading to chemoresistance by promoting STAT3-dependant EMT [172]. In melanoma, Smith et al. demonstrated that macrophage-derived TNFα secretion activate the NFκB/MITF pathway leading to resistance to MAPK-pathway inhibitors [173]. In pancreatic cancer, TAM are able to secrete pyrimidines species, such as deoxycytidine, which compete with gemcitabine and induce chemoresistance [174]. TAM also release 14-3-3ζ protein, which interacts with AXL and induces the pro-survival pathways leading to resistance to chemotherapies [175]. Additionally, in pancreatic cancer, TAM induce EMT leading to resistance to gemcitabine [176]. Similar to CAFs, TAM can confer resistance via exosomal secretion of miRNA. Indeed, Zheng et al. demonstrated that exosomal release of miR-21 by TAM M2-polarised can mediate cisplatine resistance in gastric cancer cells [177]. Maynard et al. analysed the tumour micro-environment from tumour biopsies in therapy-naïve (TN) tumours, residual disease stage (RD), and progressive disease stage (PD) [105]. The immune composition within RD was the most dissimilar from the other two stages. T cells comprised a larger fraction of all immune cells within the TME at RD compared to TN and PD, whereas macrophage infiltration showed the inverse pattern, with a decrease in macrophages at RD compared to TN and PD. They also observed fewer dysfunctional T cells than in TN or PD. Altogether, these results described a more inflammatory phenotype at RD state, hallmarked by the infiltration of T cells and decreased infiltration of immunosuppressive macrophages. The authors suggest that RD may offer a more favourable window of opportunity to introduce novel TME target-based therapies at the residual disease stage [105].

Other members of the tumour micro-environment can drive resistance to anticancer drugs, such as vasculature or oxygen levels in the tumour [67,178].

## 4. Therapeutic Perspectives

Growing evidence suggests that DTPs constitute a reservoir of cells from which both genomic and non-genomic alterations could emerge to promote the emergence of fully proliferative resistant clones. Although the existence of DTPs in patients has yet to be established, several clinical strategies are currently being tested. Combination of EGFR-TKI with immune checkpoint inhibitors (ICIs), such as durvalumab or nivolumab, do not appear to be a good option for patients due to a high rate of toxicity [179,180]. Other attractive treatment strategies could include: (1) to maintain the tumour cells in a DTP state, (2) to eradicate the DTP population by targeting specific regulators, or (3) to prevent the establishment of DTPs (Figure 5).

First, as the DTP population has been described as a heterogeneous population of slow-cycling, pseudo-dormant cells with a rare sub-population of cycling persisters [72], one option would be to eradicate specifically the dividing cells while maintaining the rest of the population in a dormant state. The use of chemotherapy in combination with EGFR-TKI may efficiently kill the proliferative cells, while sustaining the rest of the tumour cells in dormancy. A clinical trial investigated the efficacy of gefitinib combined with chemotherapy compared with gefitinib alone, and showed an improved PFS in the combination group, suggesting that the combination delayed the relapse, but did not fully prevent it [181]. Localised radiotherapy with the continuation of targeted therapy is also tested to impair the proliferative capacities of early escapers (NCT03256981). More targeted approaches could be considered to specifically target cycling cells at the DTP stage, as shown in NSCLC cell lines treated with EGFR-TKI and THZ1, a CDK7/12 inhibitor, which efficiently increases cell death and prevents relapse in vitro [123]. Similarly, the inhibition of Aurora Kinase A or B reduces cell proliferation under EGFR-TKI and increases apoptosis [124,182]. Alisertib, an Aurora Kinase A inhibitor is currently being clinically tested in combination with EGFR-TKI (NCT04085315).

Targeting the molecular and phenotypic hallmarks of DTPs could offer a relevant approach to eradicate the population. As described in the first section, DTPs are slow-proliferative, pseudo-dormant, senescent-like cells that are able to escape apoptosis upon drug treatment. Hata et al. showed that the combination of EGFR-TKI and navitoclax, a BCL-2 and BCL-x inhibitor, significantly increased apoptosis and decreased EGFR-TKI resistance [46]. This combination has been tested in a clinical trial on advanced NSCLC patients (NCT02520778) that showed that oral combination therapy with navitoclax and osimertinib was safe, but comparative studies are required to confirm its efficacy [183] (NCT02520778).

The inhibition of pathways that enable the cells to proliferate despite the drug could also be targeted to kill the population of DTPs. AXL inhibition, for example, is being tested in clinical trials (NCT02424617) as AXL inhibition delays the emergence of resistance in vitro and in vivo [41,55,127,128,129]. Similarly, the inhibition of YAP/TEAD, Wnt/b-catenin, AKT/mTOR, and FGF2/FGFR1 pathways could potentially kill the DTP population [75,95,184,185,186]. Cancer cell plasticity is also a hallmark of DTPs that enables them to reprogram themselves to adapt to their new environment, and some drugs targeting the cellular epigenome have been shown to efficiently eradicate NSCLC DTPs. For example, KDM5A inhibitor, in combination with EGFR-TKI, decreases the number of persistent cells [116]. As expected, the inhibition of IGF1R, which normally promotes H3K4demethylation by KDM5, also eliminated persistent PC9 cells [47,130]. Similarly, histone deacetylase inhibitor led to the ablation of DTPs, potentially via the disruption of heterochromatin over LINE-1-repetitive elements [113]. However, clinical trials with vorinostat, an HDAC inhibitor, revealed that even though the combination of treatment with EGFR-TKI was well tolerated, there was no significant improvement of the progression-free survival [187,188]. As metabolic reprogramming is also a key process in the establishment of DTPs, targeting such pathways may drive the elimination of DTP cells. For instance, Viswanathan el al. demonstrated that DTP survival is dependent on the lipid-peroxidase GPX4 pathway, which protects from ferroptosis-mediated cell death [90]. The enzyme inhibition induces ferroptotic death of DTPs in vitro and prevents relapse in mice models [97]. Metformin, an antidiabetic drug, has also shown synergistic activity when combined with EGFR-TKI [189], and association is currently assessed in clinics [190,191].

Finally, targeting altered tumour micro-environments represents an alternative therapeutic strategy to indirectly impact DTP survival. This strategy may be all the more interesting as it does not directly induce stress on DTP cells themselves, and thus it may prevent the canonical stress-induced reprogramming that pushes the cells towards resistance. For instance, bevacizumab, an antiangiogenic agent, has been tested in metastatic NSCLC patients in combination with EGFR-TKI. Although this combination was previously shown to prolong the progression-free survival, this last study showed that it did not extend overall survival [192], and therefore further studies are under investigation (NCT03133546).

Importantly, monitoring minimal residual disease (MRD) in patients to assess the efficiency of these potential therapeutics is essential, but remains a challenge. Liquid biopsies are an attractive alternative for cancer diagnostics and drug-response monitoring as they are minimally invasive and accessible, unlike tissue biopsies. Blood-based biopsies are mostly used to analyse circulating tumour DNA (ctDNA), which is the fragmented DNA released by cancer cells into the blood stream. The detection of genomic alterations on ctDNA is possible using next-generation sequencing (NGS) platforms or digital droplet PCR (ddPCR) and can be monitored throughout the treatment course. Interestingly, it can serve as a potential biomarker for the early detection of resistance mechanisms during targeted therapies even earlier than with radiological images [193,194,195,196,197,198]. Circulating tumour cells (CTCs) can also be detected in liquid biopsies. As they are released by the primary tumour into the bloodstream, they may be analysed as a direct proxy of the tumour at different stages of treatment, and their characterisation can provide insight into the molecular evolution of MRD. In NSCLC, we are conducting a clinical trial, LUNG-RESIST (NCT04222335), which aims to evaluate the feasibility of monitoring osimertinib drug tolerance via the monitoring of mutations in ctDNA and the phenotypic and transcriptional characterisation of CTCs during treatment. The objective is to uncover the molecular mechanisms involved in drug tolerance and resistance.

## 5. Conclusions

Resistance to targeted therapies can emerge from a subpopulation of drug-tolerant persister cells. At the molecular level, DTPs present a slow-cycling phenotype with phenotypic plasticity and display a reprogramming of cellular networks to escape drug-induced cell death. In this review, we discuss the molecular mechanisms involved in early steps of resistance to targeted therapies in non-small-cell lung cancer. Interestingly, common mechanisms of drug tolerance have been identified in other oncogenic contexts treated with various anti-cancer drugs. The studies presented in this review have led to a better characterisation of the molecular pathways underlying DTP establishment and maintenance. However, further investigation and robust preclinical models have yet to be developed to clinically target the DTPs and prevent the emergence of resistances.

## Figures and Tables

**Figure 1 cancers-14-02613-f001:**
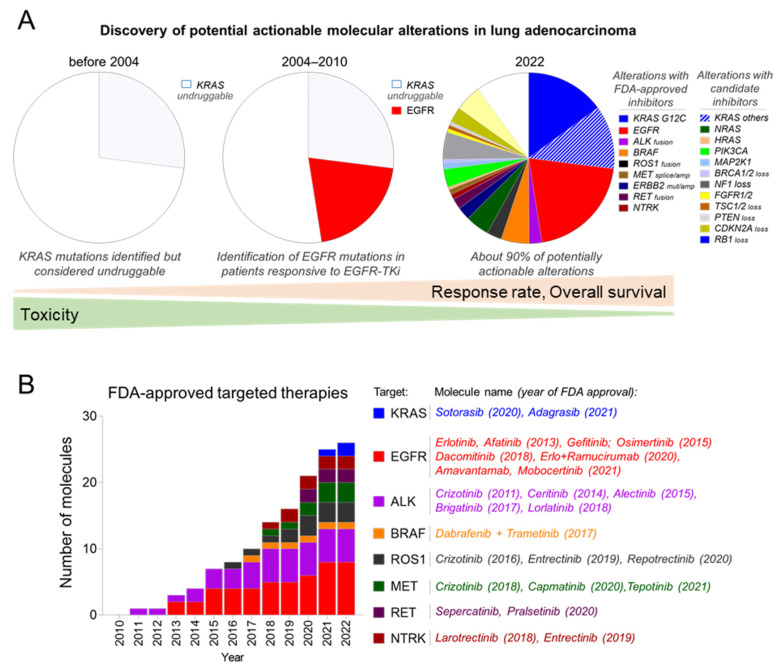
Evolution of identification of actionable alterations and appropriate targeted therapies in the standard of care in advanced non-small-cell lung cancer (NSCLC). (**A**) The discovery of oncogenic drivers in NSCLC led to the development of more personalised medicine and improved response rate and overall survival over the years. (**B**) Increasing number of FDA-approved drugs for treatment of oncogene-addicted NSCLC.

**Figure 2 cancers-14-02613-f002:**
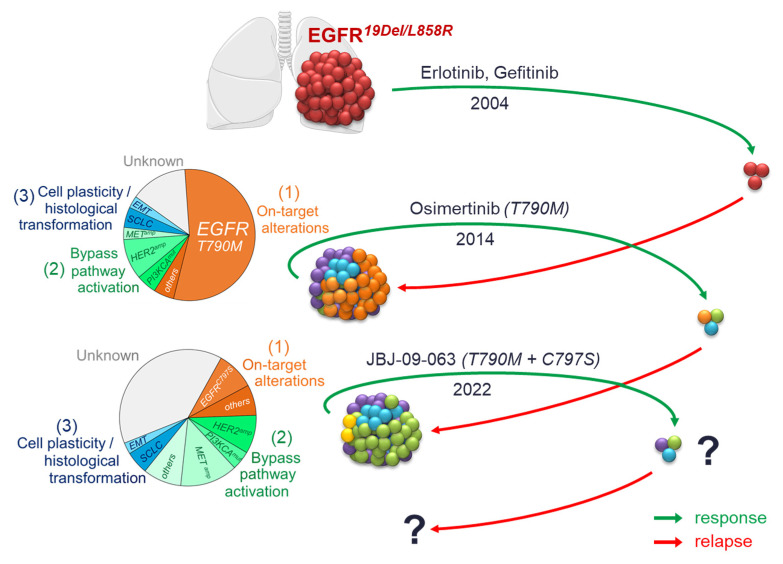
The vicious circle of resistances to EGFR-TKI in NSCLC. Although new targeted therapies are designed to counteract the resistance mechanisms developed with previous-generation EGFR-TKI, similar resistance inevitably emerges, resulting in little improvement to overall survival in patients.

**Figure 3 cancers-14-02613-f003:**
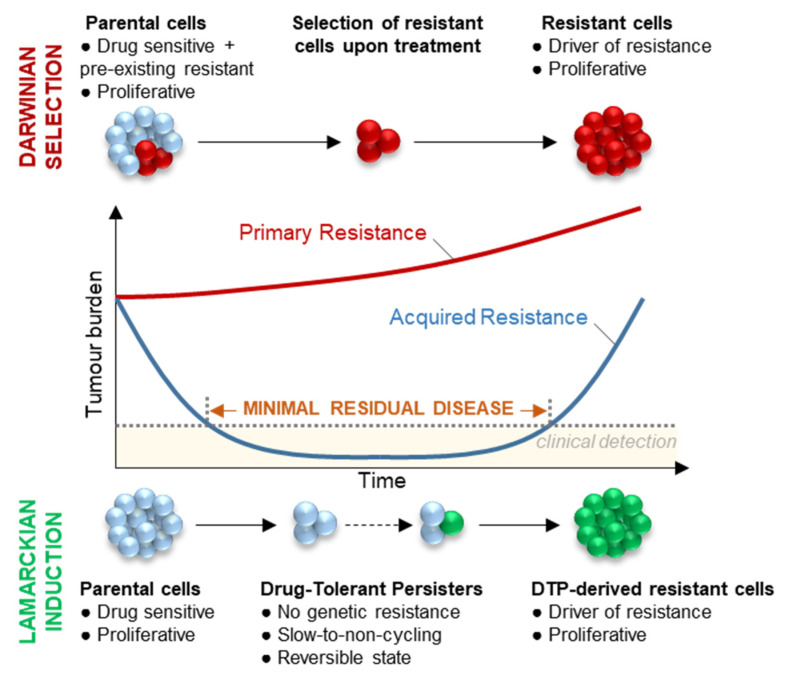
Schematic representation of the clinical response to targeted therapy in NSCLC patients. Primary resistance (red line) is characterised by a lack of initial response and a fast progression. Most patients display an initial response (blue line), with sometimes an apparently complete clearance of the tumour, although minimal residual disease persists and ultimately causes relapse in the vast majority of patients. Two hypotheses might explain the appearance of resistance for both primary and acquired resistance and might even co-exist within the same tumour. The Darwinian model suggests that, before the treatment, a population of sensitive cells (blue) co-exist with resistant cells (red) in treatment-naïve tumours, and pre-existing resistant cells are selected during treatment. The Lamarckian model suggests that a rare subpopulation of cells survives under treatment as the drug-tolerant persisters (DTPs). DTP cells exhibit non-genetic mechanisms of tolerance, slow-cycling capacities and a reversible phenotype, and could mimic the minimal residual disease phase observed in patients. DTP may eventually acquire drug-resistant mechanisms (green) and regain proliferative capacities. To study the minimal residual disease, several models are being used: most often cells treated with targeted therapies; xenograft models implanted in treated mice; and very rarely using biopsies from patients under treatment.

**Figure 4 cancers-14-02613-f004:**
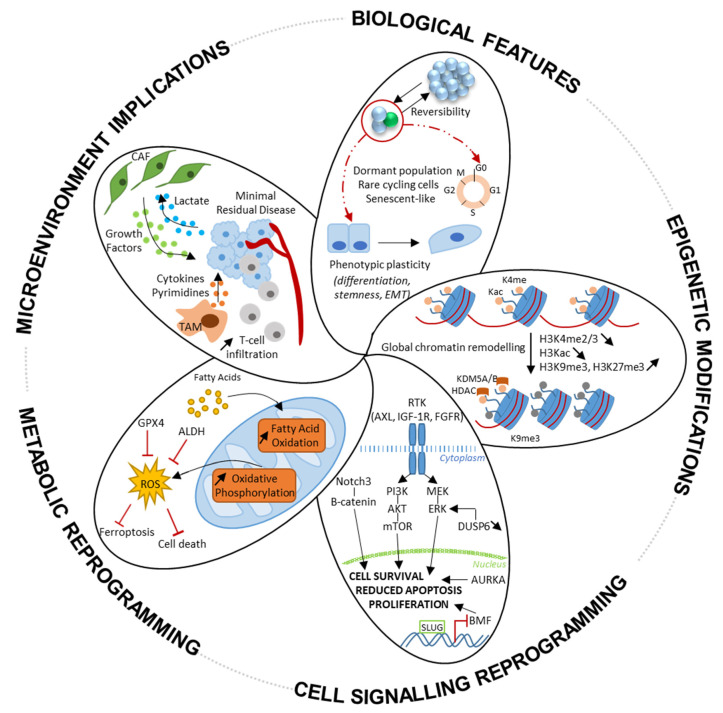
Hallmarks of drug tolerance. Drug-tolerant persisters (DTPs) share biological features, such as a reversible and plastic phenotype, slow-cycling capacities, and a senescent-like phenotype. The molecular mechanisms that drive the establishment and the maintenance of DTPs are a global chromatin remodelling with epigenetic modifications, the activation of anti-apoptotic and alternative cell-survival signalling pathways, metabolic reprogramming, and micro-environment hijacking.

**Figure 5 cancers-14-02613-f005:**
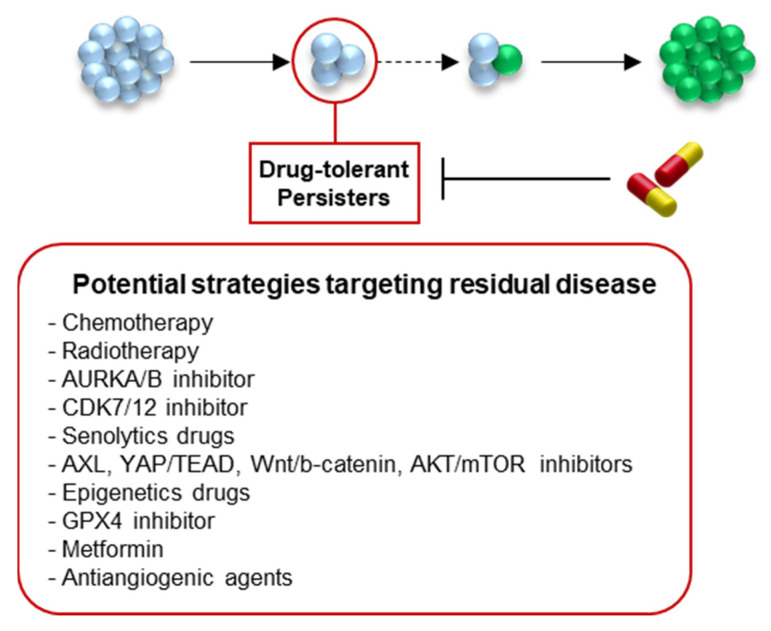
Therapeutic perspectives to improve targeted therapies outcome un lung cancer. Potential strategies to prevent the relapse include the maintenance and stabilisation of the DTP stage, the eradication of the DTP population after their establishment by targeting their key regulators, or the impediment to DTP generation. Several clinically relevant strategies are being tested.

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
