# Peer review of "Early Steps of Resistance to Targeted Therapies in Non-Small-Cell Lung Cancer"

_cancers, 2022, doi:10.3390/cancers14112613_

Round 1

Reviewer 1 Report

I went through the manuscript entitled as “Early steps of resistance to targeted therapies in lung cancer” by Drs Calvayrac and co-workers.

In general, the manuscript is well written and logically clear, which covers the important topic of how molecular resistance and clinical recurrence are taking place in the course of lung cancer treatment.

The below are the suggestions authors may take into consideration:

#1. In lung cancers NSCLC and SCLC are quite different in terms of the genetical/pathological back ground so that the treatments for NSCLC vs.SCLC are quite diverged each other. The manuscript appeared to be primarily focusing on the molecular feature of NSCLC. I would suggest authors to clearly state this point , perhaps Fig 1A or possibly in the title.

Authors state that histological transformation to small-cell lung cancers from other types of lung cancers ( line 76~77, 239~241). I feel this conversion from NSCLC to SCLC possibly be quite unusual situation, which need to provide the reference(s).

The logic of the paragraph ( lines 175~190) may not be clear to follow in my view, which may need clarification some how.

Authors discuss “liquid biopsy” as one of the method for the early detection of cancer recurrence. How “liquid biopsy” could contribute to identify the drug resistance may not clear.  I wonder if these strategies to detect fragmented DNA could help to identify the molecular nature of alteration for the drug resistance. If, so may better to provide exact reference.

In the section of “Metabolic reprogramming”, authors may add the possibility of the involvement of autophagy as underground mechanisms, since it is well established that autophagy plays a role for the regulation of metabolic process.

One of the most recent topic of cancer treatment is PD-1/PDL1 for the immunotherapy in particular advanced/ recurrent/resistant lung cancers. Authors are encouraged to discuss about PD-1/PDL1 to some extent in this review for example perspective section.

I certainly admit that most of the Figures are quite nicely presented, however, I also encourage authors to be clearer the details of the Figures carefully.

For example, Fig 1A: It is not clear that how the number of “21%” were calculated in  potentially actionable alterations, what the 15%, 12%, etc. represents---.

Fig 1B is focusing the EGFR mutation, but not clear what authors are intending to show here.

Fig 2: It is not clear what ( the plates/ mice/ lung tumor in humans) mean for the minimal residual disease, which are underneath. 

Fig 3. The figure is a little too complex and far into the details in my view.  The five arms may not correlated with the sub heading of  the section “ 3. Mechanisms of Tolerance”. The molecular mechanisms illustrated in five section appear to be too complex, which may not be necessarily proved to take place in the course of cancer treatment, in particular lung cancer.

Fig 4.   How “chemotherapy and irradiation” can be the strategies for targeting residual disease may not logically clear, as the manuscript primarily focusing on the topic of “EGFR-TK1”. It is also not certain “are the authors intend to indicate the capsule/medicine for chemotherapy/ irradiation”.

PD-1/PDL1 also be one of the alternative strategy for overcoming the recurrent stage of lung cancer in these days.

Reviewer 2 Report

Authors provide an elegant revision on a key and interesting topic such as early resistance to targeted therapy in NSCLC. The work is well-organized and figures are clear and of high quality. Only a few aspects should be addressed in addition. For completeness among mechanisms of acquired resistance some should be added such as SMO gene amplification (cfr. PMID: 26124204), together with LKB1-AMPK positive NSCLCs for which some treatment strategies have been evaluated (cfr. PMID: 26673006, 25242667). 
